# l-Carnitine and Alpha-Lipoic Acid Fail to Improve Anaerobic and Aerobic Performance in Trained Cyclists Despite a Reduction in Blood Lactate Concentration

**DOI:** 10.3390/nu17132227

**Published:** 2025-07-04

**Authors:** Alejandro de Rozas, Juan-José Pérez-Díaz, José Joaquín Muros, Cristóbal Sánchez-Muñoz, José-Ángel Rufían-Henares, Mikel Zabala, José-Antonio Salas-Montoro

**Affiliations:** 1Department of Physical Education and Sport, Faculty of Sport Sciences, University of Granada, 18071 Granada, Spain; alejandrodrg@correo.ugr.es (A.d.R.); csm@ugr.es (C.S.-M.); mikelz@ugr.es (M.Z.); salasmontoro@ugr.es (J.-A.S.-M.); 2Department of Didactics of Body Expression, University of Granada, 18071 Granada, Spain; jjmuros@ugr.es; 3Department of Nutrition and Food Sciences, Institute of Nutrition and Food Sciences, Biomedical Research Center, University of Granada, 18071 Granada, Spain; jarufian@ugr.es

**Keywords:** Acetyl-l-Carnitine, alpha-lipoic acid, aerobic performance, anaerobic performance, maximal aerobic power, blood lactate, trained cyclists, fatigue resistance

## Abstract

**Background/Objectives**: This study aimed to evaluate the effects of four weeks of combined Acetyl-l-Carnitine and alpha-lipoic acid (ALA) supplementation on anaerobic and aerobic performance and fatigue resistance in trained cyclists, hypothesizing improvements in maximal aerobic power (MAP), Wingate test performance, and reduced lactate accumulation. **Methods**: In a double-blind, randomized trial, 41 male trained cyclists (age: 36 ± 12 years; MAP: 4.35 ± 0.60 W·kg^−1^) were assigned to a supplement group (SUP, *n* = 19; 1200 mg/day Acetyl-l-Carnitine, 300 mg/day ALA, 1.1 mg Vitamin B1, 2.5 µg Vitamin B12) or placebo group (PLA, *n* = 22) for four weeks. Performance was assessed pre- and post-intervention via counter-movement jumps (CMJs), Wingate tests (WG_1_, WG_2_), and a graded exercise test (GXT). Blood lactate ([La^−^]) was measured post-Wingate. A three-way mixed ANOVA analyzed Wingate performance (session, order, and group), and a two-way ANOVA assessed MAP and fatigue effects. **Results**: MAP increased by 3.4% (314 ± 32 W to 324 ± 37 W; *p* = 0.005) with no group interaction (*p* = 0.457). Wingate peak power showed main effects for order (*p* < 0.001) and session (*p* = 0.011) but no group interaction (*p* = 0.676). SUP reduced [La^−^] by 1.5 mmol·L^−1^ post-WG_2_ in POST (*p* = 0.049). No significant group differences were found for CMJ or fatigue metrics. **Conclusions**: Four weeks of Acetyl-l-Carnitine and ALA supplementation did not enhance aerobic or anaerobic performance in trained cyclists, despite reducing blood lactate after high-intensity exercise, suggesting no ergogenic benefits.

## 1. Introduction

A nutritional supplement composed of alpha-lipoic acid (ALA), Acetyl-l-Carnitine, and vitamins B1 and B12 was used. In a previous study, it was found that the combination of these bioactive compounds might be beneficial for the prevention of oxidative stress in an in vitro intervention [1]. Furthermore, it was concluded that the ingestion of these supplements at a dose of 2250 mg/day (composed of 1200 mg/day of Acetyl-l-Carnitine and 300 mg/day of ALA) would be enough to reach high antioxidant activity, avoiding side pro-oxidant effects [1].

l-Carnitine facilitates the transport of long-chain fatty acids into the mitochondrial matrix for β-oxidation [2]. However, no improvements have appeared in moderate-intensity exercise performance (50–79% of maximal oxygen uptake [VO_2MAX_]) after acute or chronic l-Carnitine supplementation [3]. The availability of free l-Carnitine in the mitochondria appears to be limited during high-intensity exercise when the glycolytic flux and the amount of pyruvate and Acetyl-CoA increase, with the latter being a negative modulator of pyruvate dehydrogenase. Under these conditions, carnitine binds to excess Acetyl-CoA and forms Acetyl-Carnitine, which reduces Acetyl-CoA levels and allows the continuation of glycolysis while reducing blood lactate (La^−^) accumulation [4]. Previous studies have shown that l-Carnitine supplementation results in less La^−^ accumulation after high-intensity exercise using chronic doses of l-Carnitine for periods of 9 to 24 weeks [5,6,7,8].

Supplementation with ALA has been found to improve intrinsic antioxidant systems (e.g., intracellular glutathione levels or vitamin E) and scavenge reactive oxygen species (ROS) [9]. Moreover, ALA supplementation may reduce inflammatory biomarker concentrations such as C-reactive protein (CRP), Interleukin 6 (IL-6), and Tumor Necrosis Factor alpha (TNF-α) without any evidence of a non-linear association with the dose or duration of the trial [10]. Studies investigating these benefits on physical performance have found that 600 mg/day for eight days reduced oxidative damage [11] and 600–1200 mg/day for eight to ten days improved inflammatory responses after exercise [12]. However, there is insufficient supportive evidence to recommend ALA supplementation to athletes [13].

Previous studies have focused on the combined effects of Acetyl-l-Carnitine and ALA as mitochondrial-enhancing agents on the treatment of cognitive decline [14], bipolar depression [15], and disease-derived peripheral neuropathies [16]. Current recommendations of ALA and l-Carnitine supplementation are limited to patients with mitochondrial dysfunction or neurodegenerative diseases, suggesting that this population could benefit the most from its effects [17]. Considering the physiological mechanisms of l-Carnitine and ALA on the organism separately, the combination of both may exert a positive effect on exercise performance. Although l-Carnitine may increase energy production, it may also increase the production of ROS, meaning that ALA might be an excellent complementary agent to l-Carnitine for enhancing mitochondrial metabolic activity without causing an additional burden of oxidative stress, as previously demonstrated in rats [18].

Therefore, the main objective of this study was to determine the effects of four weeks of oral supplementation (a combined dose of 1200 mg/day of Acetyl-l-Carnitine, 300 mg/day of ALA, 1.1 mg of Vitamin B1, and 2.5 µg of Vitamin B12) on anaerobic performance, aerobic performance, and the ability to maintain performance under fatigue conditions. For the purposes of this study, anaerobic performance refers to peak power (P_MAX_) and average power output (P_AVG_) measured during a Wingate Anaerobic Test (WG), reflecting the capacity for high-intensity, short-duration exercise primarily reliant on anaerobic energy systems. Aerobic performance is defined as the maximal aerobic power (MAP) achieved during a graded exercise test (GXT), indicating the capacity for sustained exercise supported by aerobic metabolism.

To our knowledge, no previous studies have investigated the combined effects of Acetyl-l-Carnitine and ALA on exercise performance in athletes. While prior research has evaluated the individual effects of Acetyl-l-Carnitine and ALA on athletic performance, demonstrating benefits such as reduced lactate accumulation and oxidative damage, their combined administration has only been explored in clinical populations. This study addresses this gap by examining the synergistic potential of these mitochondrial-enhancing agents in trained cyclists, offering novel insights into their application for enhancing anaerobic and aerobic performance in a healthy athletic population.

It can be hypothesized that the supplement could exert a positive effect on high-intensity physical performance by improving maximal aerobic power (MAP) and power output in a Wingate Anaerobic Test (WG) and reducing power output decrement under fatigue conditions. In addition, it is expected that [La^−^] will be reduced after high-intensity exercise.

## 2. Materials and Methods

### 2.1. Participants

Initially, forty-six trained male cyclists and triathletes were recruited to voluntarily participate in this study. The inclusion criteria were as follows: 18 years of age or older, at least two years of regular cycling training, and three days of cycling training per week. Exclusion criteria were the presence of heart disease, metabolic disorders such as obesity (body mass index [BMI] > 30 kg·m^−2^) or diabetes, obstructive pulmonary disease, epilepsy, being on β-blocker therapy and/or medication that may alter cardiovascular function, hormone therapy, a current smoking habit, or the use of antioxidant supplements.

Three participants dropped out during the study due to personal reasons or injury unrelated to the intervention. Apart from that, there was a loss of data for two other participants due to problems with the cycle ergometer, and complete data were not obtained for them. Therefore, forty-one participants completed the study (Table 1).

All participants were informed of the procedures of the study and signed a statement of informed consent. This study was performed in accordance with the ethical principles of the Declaration of Helsinki and was approved by the Local Ethical Committee of Research (1080/CEIH/2020).

### 2.2. Experimental Design

The study had a double-blind longitudinal design, which was counterbalanced and randomized. Participants were randomly assigned to two groups—the supplement group (SUP) or the placebo group (PLA)—using a random number generator. The SUP group took the ergogenic aid for four weeks, while the control group took the placebo (rice starch) for the same period (Figure 1A). The capsules for both groups were visually identical. Participants were instructed to consume three capsules a day for the duration of the study, one before each meal (breakfast, lunch, and dinner). The participants continued with their usual training and diet throughout the duration of the study, with training volume monitored to assess potential differences between groups. Likewise, a nutritional record was kept to verify that no changes occurred in the dietary intake of Acetyl-l-Carnitine and ALA.

### 2.3. Experimental Procedures

The participants visited the laboratory to complete the performance tests on two occasions: just before starting the study (PRE) and after completing the 4 weeks of supplementation (POST). Both testing sessions were conducted on the same day of the week and at the same time. In each session, the different performance tests were carried out in the sequence below: counter-movement jump without fatigue (CMJ_1_), Wingate without fatigue (WG_1_), 10 min-recovery, a maximal graded exercise test (GXT), 5-min recovery, a WG in fatigue conditions (WG_2_), and a CMJ in fatigue conditions (CMJ_2_) (Figure 1B).

An OptoGait (Microgate Srl, Bolzano, Italy) was used to measure CMJ height through flight time. For both CMJ_1_ and CMJ_2_, participants performed two attempts, with the best of the two recorded in each case. Both WGs were performed in a cycle ergometer (Wattbike Pro, Nottingham, UK), starting from a standstill after a countdown. The WG consisted of a 30 s all-out effort, aiming to achieve the highest P_MAX_ as quickly as possible, followed by producing the maximum possible power at each moment until the end of the test. [La^−^] was measured 3 min after each WG using a drop of capillary blood from the lower part of the right earlobe (Lactate Pro 2, Arkray, Kyoto, Japan). A graded exercise test (GXT) was performed on the other cycle ergometer (Phantom 5, CycleOps, Madison, WI, USA). The GXT started with a 2 min warm-up pedaling at 100 W, followed by step increases of 25 W every two minutes until exhaustion, with the goal of achieving the highest possible MAP. The GXT ended when the cyclist voluntarily decided to stop because they were unable to maintain the required intensity. During the GXT, [La^−^] and the rate of perceived exertion (RPE) using the Borg CR10 scale were measured at the end of every step. Heart rate (HR) was monitored throughout the entire session using a chest strap (H10, Polar, Kempele, Finland).

### 2.4. Data Analysis

To evaluate the performance decline produced in each session, the difference between the height of the CMJ_1_ and CMJ_2_ jumps was calculated, as well as the performance values of the WG tests. The height difference between both CMJs (ΔCMJ) was expressed as a percentage change of CMJ_2_ with respect to CMJ_1_ and was calculated as shown in Equation (1):(1)ΔCMJ%=CMJ2−CMJ1CMJ1·100

In the WG, P_MAX_ was the highest value reached, and the average power (P_AVG_) was the mean power of the 30 s. For both P_MAX_ and P_AVG_, the percentage change of the WG_2_ with respect to the WG_1_ was calculated using Equations (2) and (3):(2)ΔPMAX%=PMAX2−PMAX1PMAX1·100(3)ΔPAVG%=PAVG2−PAVG1PAVG1·100

In the GXT, MAP was the power achieved in the last completed step. If the participant was not able to complete a full step, MAP was calculated as the power achieved in the last completed step plus the proportional part of the uncompleted step [19], according to Equation (4):(4)MAP W= Last completed step W+time not completed s120 s·25 W

Power data were reported in absolute terms (W) and related to body mass (W·kg^−1^).

### 2.5. Statistical Analysis

The study variables are represented as means and standard deviations for PRE and POST. To determine the homoscedasticity of the variances, Levene’s test was applied. To analyze the normality of the data, the Shapiro–Wilk model was used. To examine the interaction effects of the session with the group in the MAP and the performance decline, a two-way repeated-measure analysis of variance (ANOVA) test was used. To assess the Wingate performance, a three-way mixed ANOVA was conducted to evaluate the effects of session (PRE-POST), order (WG_1_-WG_2_), and group (SUP-PLA), as well as their two-way and three-way interactions. Statistical significance was accepted when *p* < 0.05. Whenever a significant *p*-value was obtained, post hoc testing was performed with a Bonferroni correction for multiple comparisons. The analyses were performed using SPSS^®^ software version 21.0 (SPSS, Inc., Chicago, IL, USA).

## 3. Results

The initial data regarding the participants’ characteristics showed no significant differences between groups (age, *p* = 0.383; body mass, *p* = 0.950; height, *p* = 0.435; BMI, *p* = 0.473; MAP, *p* = 0.909). Likewise, there were no differences in weekly training volume either (*p* = 0.071).

### 3.1. Anaerobic Performance

For P_MAX_ in the WG test, there was a main effect of order (WG_1_ vs. WG_2_; F(1, 39) = 20.303, *p* < 0.001; $\eta_{p}^{2}$ = 0.342) and session (PRE vs. POST; F(1, 39) = 7.091, *p* = 0.011; $\eta_{p}^{2}$ = 0.154). However, there was no significant interaction between the factors order, session, and group (*p* = 0.676). Post hoc analysis revealed that P_MAX_ increased by a mean difference of 0.639 W·kg^−1^ in WG_2_ compared to WG_1_, as well as 0.358 W·kg^−1^ in the POST session compared to the PRE session.

For P_AVG_, there was only a significant effect of order (F(1, 39) = 16.017, *p* < 0.001; $\eta_{p}^{2}$ = 0.291), with a mean difference of 0.374 W·kg^−1^ higher in WG_1_ compared to WG_2_. There was no difference between sessions (*p* = 0.296) and no interaction between the factors order, session, and group (*p* = 0.253).

All individual power values obtained can be found in Appendix A (Table A1).

Blood [La^−^] analysis showed significant effects for order (F(1, 39) = 108.941, *p* < 0.001; $\eta_{p}^{2}$ = 0.736), the order × group interaction (F(1, 39) = 7.567, *p* = 0.009; $\eta_{p}^{2}$ = 0.162), and the order × session × group interaction (F(1, 39) = 4.132, *p* = 0.049; $\eta_{p}^{2}$ = 0.096). There were no significant differences between groups (*p* = 0.434). Nevertheless, after WG_2_, [La^−^] decreased by 1.5 mmol·L^−1^ in the POST session compared to the PRE session in the SUP group (from 15.9 to 14.4 mmol·L^−1^), while in the PLA group, it increased by 0.2 mmol·L^−1^ (from 14.2 to 14.4 mmol·L^−1^) (Figure 2).

No significant interactions were observed between session (PRE and POST), order (WG_1_ and WG_2_), and group (SUP and PLA) for HR_MAX_ (*p* = 0.831) or RPE (*p* = 0.532).

### 3.2. Aerobic Performance

The MAP increased by 3.4% over four weeks (F(1, 39) = 8.694, *p* = 0.005; $\eta_{p}^{2}$ = 0.182), from 314 ± 32 W to 324 ± 37 W (4.35 ± 0.60 W·kg^−1^ to 4.49 ± 0.61 W·kg^−1^), although no significant interaction was observed between groups (F(1, 39) = 0.566, *p* = 0.457; $\eta_{p}^{2}$ = 0.014). A two-way mixed ANCOVA, with training volume as a covariate, showed no significant interaction for MAP (*p* = 0.956), indicating that training volume did not differentially affect the supplementation’s impact on MAP changes from PRE to POST. In Figure 3, the average MAP data are shown, along with all individual data points.

Neither the HR nor the RPE differed between sessions (*p* = 0.078 and *p* = 0.426 for HR and RPE, respectively) or in the session × group interaction (*p* = 0.195 and *p* = 0.953 for HR and RPE, respectively) at the end of the GXT.

### 3.3. Performance Decline

In the WG, there were no significant differences between groups for ΔP_MAX_ (*p* = 0.468), as well as no significant Session × Group interaction (*p* = 0.542). No significant effects were found for P_AVG_ either (Group: *p* = 0.409; Session × Group interaction: *p* = 0.235). In Figure 4, the average and standard error of ΔP_MAX_ and ΔP_AVG_ data are shown, along with all individual data points.

Jump height in the CMJ showed significant effects for order (CMJ_1_ vs. CMJ_2_; F(1, 39) = 33.315, *p* < 0.001; $\eta_{p}^{2}$ = 0.461), session (PRE vs. POST; F(1, 39) = 8.625, *p* = 0.006; $\eta_{p}^{2}$ = 0.181), and the order × session interaction (F(1, 39) = 5.614, *p* = 0.023; $\eta_{p}^{2}$ = 0.126). However, there were no significant effects for group (*p* = 0.730) or interaction between the factors order, session, and group (*p* = 0.306). Post hoc analysis revealed that jump height increased by a mean difference of 0.9 cm in the POST session, while between the two jumps of each session, it decreased by a mean difference of 2.6 cm. Regarding ΔCMJ, in the POST session, the percentage loss decreased by a mean difference of 4% compared to the PRE session (F(1, 39) = 5.995, *p* = 0.019; $\eta_{p}^{2}$ = 0.133), showing no differences between groups (*p* = 0.132) or in the session × group interaction (*p* = 0.309). In Figure 5, the average and standard error of ΔCMJ data are shown, along with all the individual data points.

## 4. Discussion

To the best of our knowledge, this is the first study that has evaluated the effect of chronic oral supplementation with Acetyl-l-Carnitine and ALA on physical performance in trained cyclists. The purpose of this study was to determine the effects of long-term (four weeks) oral supplementation with this combined supplement on aerobic and anaerobic performance under fresh and fatigue conditions in trained cyclists. The main results obtained from this study show that four weeks of supplementation with this combined formula does not improve any of the performance variables measured in this study, including P_MAX_ and P_AVG_ in the WG, MAP, or jump height. Although a statistically significant reduction in [La^−^] of 1.5 mmol·L^−1^ was observed post-WG_2_ in the SUP group (*p* = 0.049), this biochemical change did not translate to improvements in performance or fatigue resistance, suggesting limited practical significance for athletic outcomes. These results showed that four weeks of supplementation with the indicated doses did not improve cycling performance, which could suggest that the supplement has no effect on performance or that the dose or duration of supplementation was insufficient to enhance performance. Therefore, more research is required to establish the use of this combined supplement evidence in this population.

The dose of the combined supplement used in our study is in line with the results of the Lopez-Maldonado et al. study [1], in which it was found that the antioxidant capacity of this combined formula increased in a dose-dependent manner, reaching its maximal potential at the dose of three capsules per day. Although they suggested an antioxidant effect in an in vitro intervention, no effects were found in antioxidant capacity in humans after the ingestion of six pills/day of this supplement for 30 days. However, the 300 mg/day ALA dose is lower than the 600–1200 mg/day typically used in exercise-related studies [11,12], although the duration of the intake exceeded the 8–10 days reported in these studies, which may have limited its ergogenic potential. Similarly, while the 4-week supplementation period is not uncommon, prior studies reporting performance improvements with Acetyl-l-Carnitine or ALA often extended over 9 to 24 weeks [5,6,7,8]. The lower dose and shorter duration may explain the absence of performance improvements in our study, suggesting that the supplement may not be ineffective but rather underdosed or insufficiently timed to elicit measurable effects. Since this is the first study that has evaluated the combined effect of Acetyl-l-Carnitine and ALA on physical performance in humans, we have discussed our findings with studies that have assessed the effects of either Acetyl-l-Carnitine or ALA, separately. As a limitation of the study, muscle carnitine content, oxidative biomarkers, and indicators of mitochondrial function were not measured, precluding direct validation of the proposed mechanisms involving Acetyl-CoA buffering, oxidative stress reduction, and enhanced mitochondrial function. Consequently, the ingestion of Acetyl-l-Carnitine may not be the sole factor responsible for the reduction in blood [La^−^], and the lower ALA dose and shorter supplementation period may have further limited the ability to detect performance benefits.

Few studies have been conducted in humans regarding the effect of ALA supplementation on exercise performance. Zembron-Lacny et al. [11] found no significant improvements in performance during three 10 s maximal voluntary contractions of the quadriceps after the ingestion of 600 mg/day for eight days. In another study of the same group, subjects ingested 1200 mg/day of ALA for 10 days before performing a 90 min run followed by a 15 min eccentric run at 65% VO_2MAX_. They found a positive modulation of inflammatory response (IL-6) and muscle damage (CK) biomarkers after exercise, whereas exercise performance was not assessed [12]. In our study, subjects ingested a lower daily dose of ALA (300 mg/day) for a longer period (four weeks), with no improvements in exercise performance. However, since our supplementation and testing protocols differ substantially, comparisons between results should be made with caution. Given that there is no evidence of the dose or duration of the intervention [10], more research is needed in order to establish the optimal ALA supplementation protocol.

To discuss the findings of the present study in reference to previous work, it is useful to first consider the known metabolic functions of Acetyl-l-Carnitine and their influence on exercise at different intensities. Firstly, the binding of l-Carnitine to acetyl groups through the action of carnitine acyltransferase plays a vital role in the transport of long-chain fatty acids into the mitochondria for their subsequent β-oxidation within the mitochondrial matrix. This metabolic function is a determinant in moderate-intensity exercise, during which Adenosine Triphosphate (ATP) regeneration is dependent on fatty acids and glucose oxidation. Secondly, l-carnitine has also been recognized for its crucial biological function in buffering the Acetyl-CoA/CoA ratio, preventing the accumulation of Acetyl-CoA. Elevated levels of Acetyl-CoA can hinder the activity of the pyruvate dehydrogenase complex (PDC), leading to an increase in La^−^ production. In this scenario, carnitine accepts the acetyl group and forms Acetyl-l-Carnitine. This reduces the levels of Acetyl-CoA, allowing the continuation of glycolysis during high-intensity exercise. However, this process is limited by the availability of muscle carnitine, which gradually diminishes with sustained high-intensity exercise. Consequently, muscle carnitine levels have been associated with the ability to sustain high-intensity efforts while reducing La^−^ production.

Regarding high-intensity exercise performance, prior research has found a positive effect of chronic and acute Acetyl-l-Carnitine supplementation on high-intensity exercise performance (≥80% VO_2MAX_) using chronic doses of 2 to 2.72 g/day of Acetyl-l-Carnitine for longer periods up to 9 to 24 weeks [3]. In our study, no significant improvements were found in WG_1_ or WG_2_. Wall et al. [8] found that a dose of 2.72 g/day of Acetyl-l-Carnitine in combination with 160 g/day of carbohydrates (CHO) improved work output in a 30 min test after 24 weeks (11% greater than the baseline and 35% greater than the control group), but no differences were found after 12 weeks. In another study, there was a significant improvement in P_MAX_, but not in P_AVG_, during a WG test following a dose of 2 g/day of Acetyl-l-Carnitine for 9 weeks [7]. These findings may suggest that the Acetyl-l-Carnitine dose and supplementation period of our study (1.2 g/day of Acetyl-l-Carnitine for 4 weeks) might be too short to find a significant effect on high-intensity performance. In line with our results, other studies with similar chronic supplementation protocols did not obtain improvements in high-intensity performance. Shannon et al. [20] did not find differences between groups in work output during two 3-min cycle bouts at MAP after 24 weeks of 3 g/day of Acetyl-l-Carnitine with 160 g of CHO. Smith et al. [21] found no significant interaction between groups for P_MAX_ or P_AVG_ in a WG test after eight weeks of either 1 g/day or 3 g/day Acetyl-l-Carnitine supplementation.

In our study, ΔP_AVG_ revealed a reduction in power output from WG_1_ to WG_2_ for both groups with no group interaction. However, an increase in P_MAX_ was observed under fatigue compared to fresh conditions. These findings might be due to the lack of a proper warm-up before the WG_1_. This reduction in anaerobic power output has also been reported in previous studies. Jacobs et al. [6] found that power output tended to diminish significantly across five 10 s sprint bouts, with significant differences between groups only from the third sprint onwards. However, our results are not comparable to the findings of the Jacobs et al. study since the performance tests, supplementation protocols, and populations differed substantially. On the other hand, our results show no benefit in aerobic performance. In line with our study, Broad et al. [22] did not find any differences between groups after four weeks of 2 g/day of Acetyl-l-Carnitine during a 20 km time trial following a 90 min steady-state ride at 65% VO_2MAX_. Shanon et al. [20] did not find significant differences in MAP during a GXT after 24 weeks of 3 g/day of Acetyl-l-Carnitine with 160 g of CHO/day. In contrast, Orel and Guzel [5] found an increase in running speeds corresponding to different [La^−^] between 2 and 4 mmol·L^−1^ during a GXT with the acute ingestion of 3 g or 4 g of Acetyl-l-Carnitine one hour before, suggesting a greater time to exhaustion while exercising at submaximal intensity.

Despite the short supplementation period, our results showed that Acetyl-l-Carnitine supplementation reduced [La^−^] after high-intensity exercise. In agreement with this, Jacobs et al. [6] showed a reduced blood [La^−^] accumulation after five 10 s sprints where Acetyl-l-Carnitine was ingested in a single dosage of 4.5 g of glycine propionyl-l-carnitine 90 min prior to exercise. Furthermore, a significant decrease in 3 min post-exercise blood [La^−^] was observed at week nine, but not after week six, in the experimental group following a dose of 2 g/day of Acetyl-l-Carnitine compared with the placebo [7]. Furthermore, Wall et al. [8] found that when Acetyl-l-Carnitine is ingested for 24 weeks in combination with CHO, muscle La^−^ content was reduced following exercise at 80% VO_2MAX_. Although we did not find any performance differences derived from less blood [La^−^] accumulation, these studies also revealed improvements in high-intensity exercise performance. On the other hand, prior studies had not found any difference in [La^−^] using Acetyl-l-Carnitine doses of 2–4 g/day for 7 days [23], 14 days [24], or 24 weeks [20]. Furthermore, some previous studies have indicated that l-carnitine does not provide high-intensity performance benefits during five 1 min bouts at 115% VO_2MAX_ [24] or during a GXT and a WG [21], due to the inability to significantly increase muscle carnitine concentrations by oral supplementation. This difficulty of increasing muscle carnitine content could be improved with increased insulin levels [25]. Moreover, Mielgo-Ayuso et al. [3] concluded that supplementation with Acetyl-l-Carnitine and CHO over periods of 9 to 24 weeks could be beneficial in improving performance in GXT, 10 s sprints, WG, and all-out tests, suggesting a greater effect when supplementation with Acetyl-l-Carnitine includes CHO. The increased availability of CHO during exercise increases glycolysis flux and Acetyl-CoA production through the pyruvate dehydrogenase. Under these conditions, carnitine buffers the Acetyl-CoA/CoA ratio, allowing the continuation of glycolysis and the conversion of pyruvate to Acetyl-CoA, resulting in less La^−^ accumulation. As previously shown, a decrease in the rate of fat oxidation with an increase in La^−^ production has been observed during high-intensity exercise [26], likely due to the limited availability of carnitine when exercise intensity increases. This observation should lead to paying attention to the use of carnitine as a supplement to increase the burning of fatty acids and the continuation of glycolysis during high-intensity exercise when carnitine administration is preceded by a load of CHO before exercise, to ensure that glycogen availability is not limiting exercise performance. Not having ingested CHO could have limited the Acetyl-l-Carnitine potential in our study. Moreover, it is likely that the duration of exercise performed above 75% of MAP during the GXT (6.36 ± 0.77 min) in this study was not enough to allow a plateau of acetyl accumulation and the inhibition of pyruvate dehydrogenase complex activity, which is normally observed within 10 min at 75–90% VO_2MAX_ [27]. Furthermore, the 10 min recovery period between WG_1_ and the subsequent GXT of our study is hypothesized to have potentially allowed the reversal of the Acetyl-CoA and acylcarnitine content to basal levels, as previously found [28], though this remains speculative without direct biomarker measurements. Therefore, the ability of Acetyl-l-Carnitine to buffer acetyl groups in our study would be less critical when it comes to finding improvements.

The statistically significant interaction for [La^−^] post-WG_2_ in the SUP group (*p* = 0.049), with a 1.5 mmol·L^−1^ reduction, must be interpreted cautiously due to the absence of improvements in any performance outcomes. As multiple performance variables were tested in this study, the *p*-value of 0.049, close to the 0.05 threshold, raises the possibility of a Type I error, where the observed La^−^ reduction may have occurred by chance. This suggests that the La^−^ reduction is unlikely to have practical significance for athletic performance, consistent with the overall null findings for performance metrics.

As shown in a previous longitudinal research, metabolic and neuromuscular adaptations occur following endurance training among elite cyclists, thus improving physical performance throughout the season [29]. The participants in our study were trained cyclists, so they continued training during the study period. Considering that the study was conducted during the preparatory period (January and February), when cyclists typically increase training volume and intensity to build fitness, this seasonal training effect represents a potential confounder that likely explains the observed 3.4% increase in MAP across both groups (from 314 ± 32 W to 324 ± 37 W), independent of supplementation. This likely explains the observed 4% reduction in ΔCMJ from PRE to POST sessions (*p* = 0.019), which was not group-specific (*p* = 0.309) and reflects a training-induced adaptation in fatigue resistance rather than an effect of the supplement. Training volume was monitored during the study, with no significant differences between the SUP and PLA groups (*p* = 0.071), suggesting that group-specific training adaptations were unlikely. A two-way mixed ANCOVA, with training volume as a covariate, confirmed no significant interaction for MAP (*p* = 0.956), further supporting that the MAP increase was not influenced by differential training volume effects between the supplement and placebo groups.

Lastly, the current recommendations of ALA and Acetyl-l-Carnitine supplementation are limited to patients with mitochondrial dysfunction or neurodegenerative diseases, suggesting that this population could benefit the most from its effects. Therefore, more research is needed to elucidate the potential effects of this combination in healthy subjects and the athletic population.

## 5. Conclusions

In conclusion, our trial demonstrated that supplementation with the combined formula, over a period of four weeks, was not able to improve aerobic or anaerobic physical performance in a relatively large cohort of male trained cyclists. Despite a statistically significant reduction in [La^−^] of 1.5 mmol·L^−1^ post-WG_2_ in the SUP group (*p* = 0.049), this change did not translate to enhanced performance or fatigue resistance, indicating no ergogenic benefit under the tested conditions. These findings do not support practitioners’ decisions in recommending its supplementation for athletes.

## Figures and Tables

**Figure 1 nutrients-17-02227-f001:**
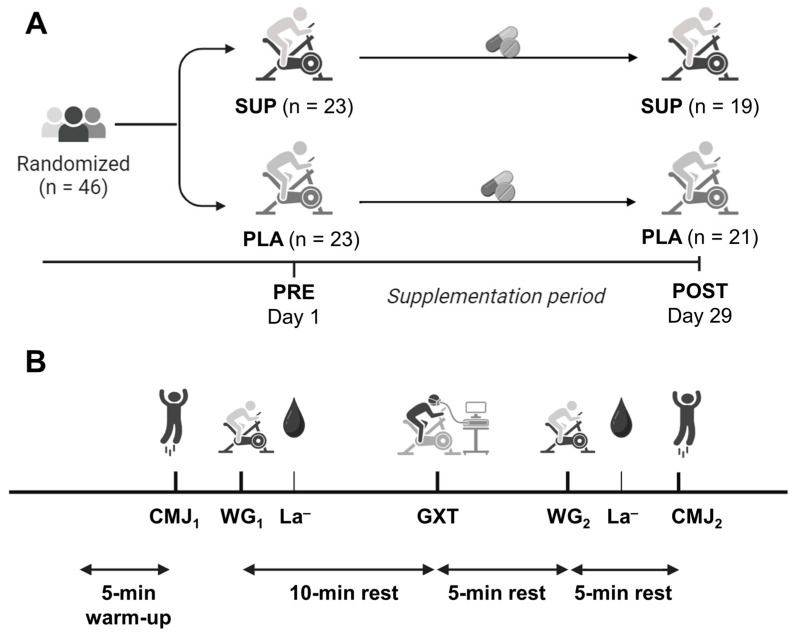
Schematic summary of the procedures carried out in the study. (**A**) Experimental design, with the supplement (SUP) and placebo (PLA) groups, as well as the timeline of the supplementation period and assessment tests; (**B**) temporal sequence of the assessment tests conducted in each of the two evaluation sessions. CMJ, counter-movement jump; WG, Wingate; La^−^, lactate; GXT, graded exercise test.

**Figure 2 nutrients-17-02227-f002:**
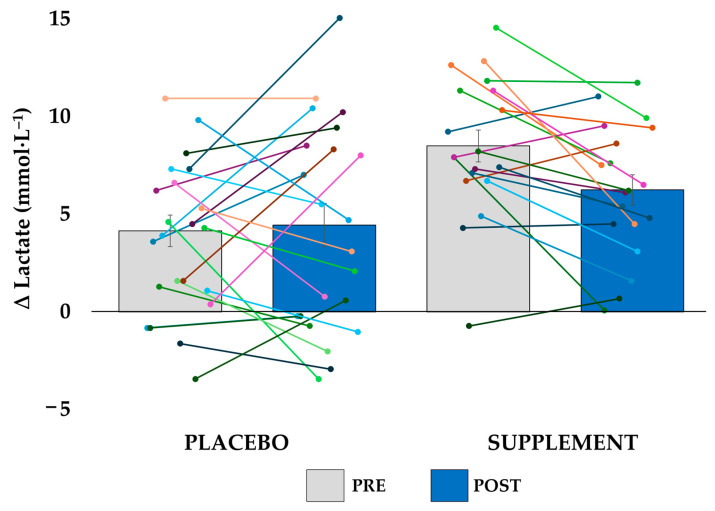
Variation in lactate concentration between both Wingate tests for the placebo and supplement groups in the pre-intervention (PRE) and post-intervention (POST) sessions. Bars represent the group averages, with error bars indicating the standard error. Individual participant data are shown as overlaid points. Positive values represent a higher lactate level in the second Wingate test.

**Figure 3 nutrients-17-02227-f003:**
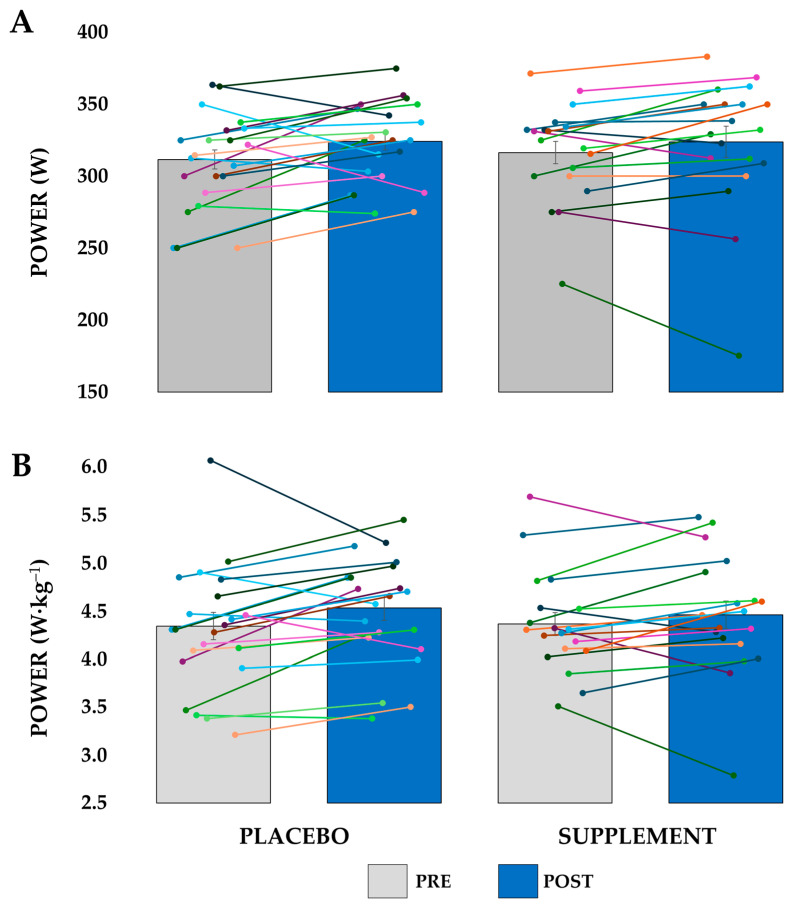
Maximal aerobic power for the placebo and supplement groups in the pre-intervention (PRE) and post-intervention (POST) sessions. Bars represent the group averages, with error bars indicating the standard error. Individual participant data are shown as overlaid points. (**A**) Maximal aerobic power in watts; (**B**) maximal aerobic power relative to body mass in W·kg^−1^.

**Figure 4 nutrients-17-02227-f004:**
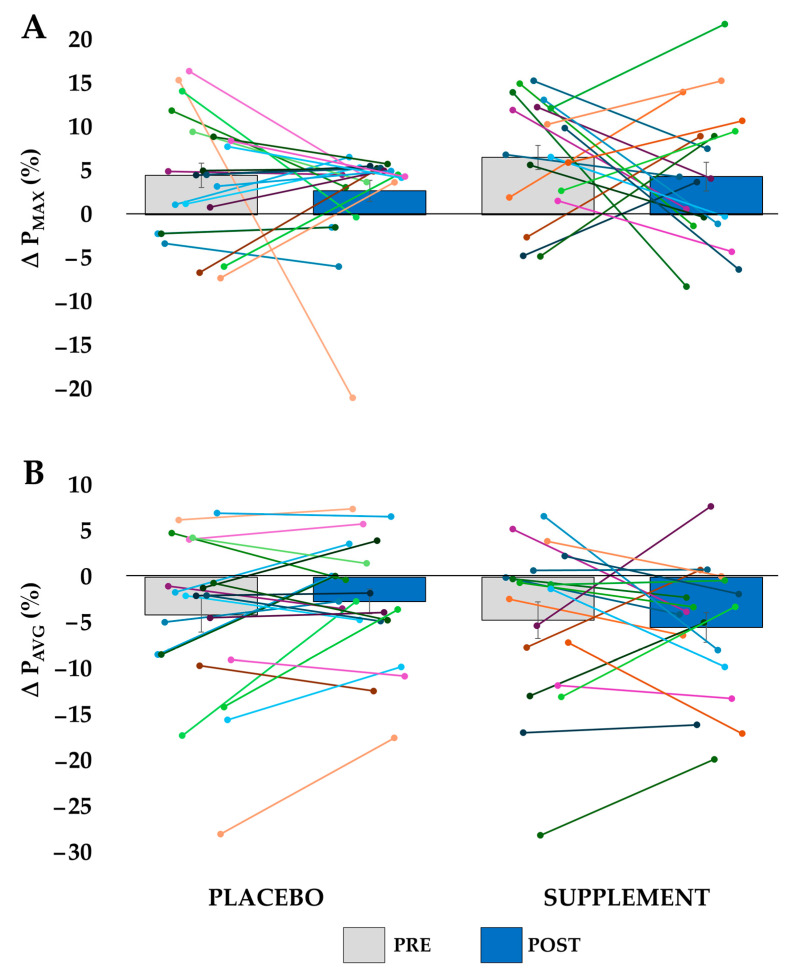
Percentage change of WG_2_ relative to WG_1_ for the placebo and supplement groups in the pre-intervention (PRE) and post-intervention (POST) sessions. Bars represent the group averages, with error bars indicating the standard error. Individual participant data are shown as overlaid points; (**A**) change in peak power (ΔP_MAX_); (**B**) change in average power (ΔP_AVG_).

**Figure 5 nutrients-17-02227-f005:**
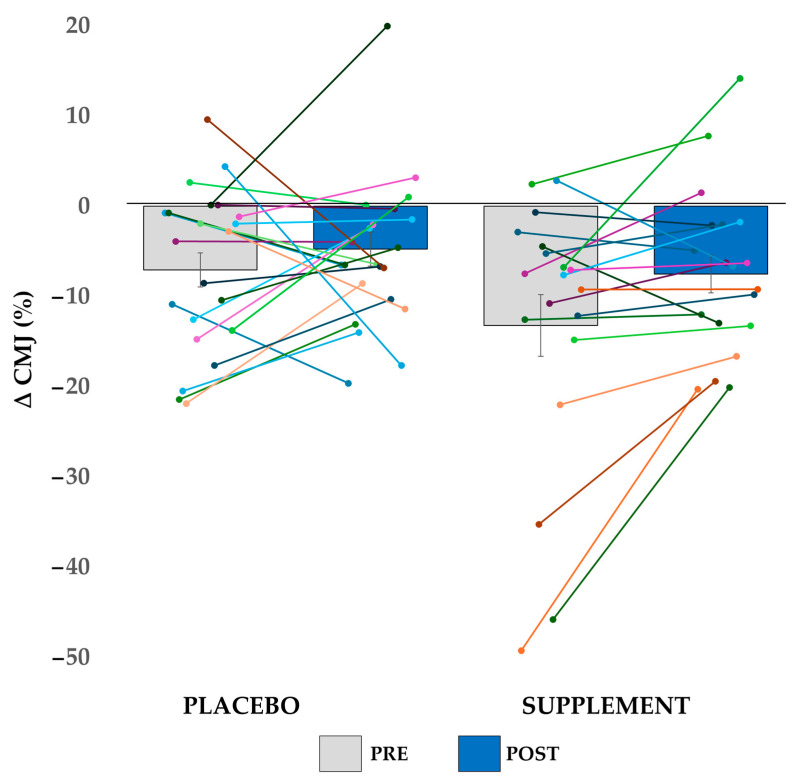
Percentage change of CMJ_2_ relative to CMJ_1_ (ΔCMJ) for the placebo and supplement groups in the pre-intervention (PRE) and post-intervention (POST) sessions. Bars represent the group averages, with error bars indicating the standard error. Individual participant data are shown as overlaid points.

**Table 1 nutrients-17-02227-t001:** Physical and performance characteristics (mean and standard deviation) of all participants in the initial study session. The values related to the counter-movement jump and Wingate refer to the first test performed.

	All Participants	Supplement Group	Placebo Group
Age (y.o.)	36 ± 12	38 ± 10	35 ± 13
Body Mass (kg)	72.9 ± 8.7	73.0 ± 8.0	72.8 ± 9.4
Height (cm)	177.6 ± 7.1	176.7 ± 8.3	178.5 ± 6.0
Body Mass Index (kg·m^−2^)	23.1 ± 2.6	23.4 ± 2.6	22.9 ± 2.6
Average training volume (h/week)	10.2 ± 3.7	9.0 ± 3.3	11.1 ± 3.9
Maximal heart rate (bpm)	185 ± 11	180 ± 12	188 ± 10
Counter-movement Jump (cm)	28.8 ± 5.4	28.9 ± 4.8	28.6 ± 6.0
Peak Power (W)	1090 ± 176	1075 ± 182	1104 ± 174
Pear Power (W·kg^−1^)	15.05 ± 2.41	14.75 ± 2.07	15.31 ± 2.69
30-s Average Power in Wingate (W)	692 ± 102	683 ± 95	701 ± 110
30-s Average Power in Wingate (W·kg^−1^)	9.56 ± 1.52	9.39 ± 1.20	7.73 ± 1.76
Maximal Aerobic Power (W)	314 ± 32	316 ± 34	311 ± 31
Maximal Aerobic Power (W·kg^−1^)	4.35 ± 0.60	4.36 ± 0.52	4.34 ± 0.67

## Data Availability

Data are available from the authors on request. Data are not publicly available due to privacy reasons.

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
