# Peer review of "l-Carnitine and Alpha-Lipoic Acid Fail to Improve Anaerobic and Aerobic Performance in Trained Cyclists Despite a Reduction in Blood Lactate Concentration"

_nutrients, 2025, doi:10.3390/nu17132227_

Round 1
Reviewer 1 Report
Comments and Suggestions for Authors
This study aimed to evaluate the effects of four weeks of combined Acetyl-L-Carnitine and alpha-lipoic acid (ALA) supplementation on anaerobic and aerobic performance and fatigue resistance in trained cyclists. As the results, four weeks of Acetyl-L-Carnitine and ALA supplementation reduced blood lactate after high-intensity exercise but did not enhance aerobic or anaerobic performance in trained cyclists. This study examined the effects of combined Acetyl-L-Carnitine and ALA supplementation on anaerobic and aerobic performance in athletes, and the reviewer believes that it will provide useful information in terms of improving athletes’ performance and reducing fatigue. However, there are several limitations in this study.
- What is novel about this study? As the authors explain in the Introduction section, there have been many published reports of benefits from Acetyl-L-Carnitine and ALA supplementation. The authors should explain how their study differs from previous studies and what makes it novel.
- The definitions of “anaerobic performance” and “aerobic performance” in this article are unclear. The reviewer recognizes that the results obtained from the Wingate Test (WG) represent anaerobic performance (Pmax and Pave), and the results obtained from a graded exercise test (GXT) represent aerobic performance (MAP). The authors should properly express the definitions of the terms in this article.
- In relation to the above, in this article, the authors defined “fatigue” as the percentage of changes in the results of WG, GXT, and CMJ. The reviewer believes that it is inappropriate to define “fatigue” as the percentage of changes in the results of each test.
- The figures in this article show the results for MAP, Pmax, Pave, and ∆CMJ. However, even though the title and purpose of this study include “blood lactate”, the results for “blood lactate” are not shown. The reviewer considers that the results of this article should also include a figure showing “blood lactate” because “blood lactate” is a very important outcome in this study.
- In relation to the above, in this study, the authors also measured heart rate (HR) and rate of perceived exertion (RPE), but the results of HR and RPE were not presented. The reviewer thinks that the measurements described in the “Methods” section should also be presented in the “Results” section.
- In this article, the authors only provide brief information on the participants’ physical characteristics in the “Methods” and “Results” sections. The reviewer believes that information about the physical characteristics of participants is very important in a human study. The authors should provide a table with information on the physical characteristics of participants in both the supplement and placebo groups.
- The explanation of WG and GXT is insufficient in the “Methods” section. The current description does not allow the reader to understand what measurements were performed. Please explain in detail so that readers of this article can easily understand.
- Appendix Table A presents the detailed results of the WG. The reviewer thinks that readers will find it easier to understand the results in this table if they are shown graphically time course rather than as numerical information.
Reviewer 2 Report
Comments and Suggestions for Authors
General Comments
Although the study presents a comprehensive assessment of performance variables (aerobic, anaerobic, and fatigue-resistance), I am concerned that the discussion does not sufficiently emphasize the essentially null nature of the primary outcomes. While the observed reduction in post-exercise lactate is statistically significant, I did not find evidence that this translated into any measurable performance enhancement. Thus, in my view, the conclusion that the supplement offers “limited ergogenic benefit” is overly cautious—one could more directly state that no ergogenic benefit was found under the tested conditions.
Throughout the manuscript, the authors refer to theoretical mechanisms involving Acetyl-CoA buffering, oxidative stress reduction, and mitochondrial function. While these references are supported by prior literature, I believe the study would benefit from a more explicit acknowledgment that none of these mechanisms were empirically assessed in the current trial. For instance, muscle carnitine content, oxidative biomarkers, or indicators of mitochondrial function were not measured. Without such data, it becomes difficult to draw any conclusions about the physiological relevance of the lactate response or to validate the proposed mode of action.
I found the choice of supplement dose—particularly the 300 mg/day of ALA—to be lower than what is typical in exercise-related studies, where doses of 600–1200 mg/day have been used. Likewise, while four weeks of intervention is not uncommon, prior studies reporting performance improvements typically extended over 9 to 24 weeks. I believe this issue warrants a more detailed discussion, particularly in light of the null results. As a reader, I am left wondering whether the supplement was ineffective or merely underdosed and undertimed.
The statistical analyses are appropriately conducted, and figures (notably Figures 2–4) offer transparent visualizations of individual and group-level trends. However, despite a statistically significant reduction in lactate of 1.5 mmol/L post-WG2 (p = 0.049), I would question whether this result has practical significance. This subtle biochemical change did not appear to impact fatigue resistance, peak or mean power, or jump height. I suggest that the authors temper their interpretation of the lactate result and avoid implying performance relevance where none is observed.
The authors note that testing occurred during a preparatory period of training (January–February), when athletes may have experienced natural increases in fitness. Indeed, the rise in MAP across both groups (314 ± 32 W to 324 ± 37 W) could plausibly be explained by training adaptation rather than supplementation. In my opinion, this seasonal confounder should be explicitly acknowledged and possibly tested via interaction analysis if training volume data are available.
Specific Comments by Line Numbers
- Lines 29–32: I would recommend stating more assertively that no improvements in performance were observed, despite the reduction in blood lactate. The current phrasing underplays this null finding.
- Lines 87–99: Given that participants are described as “highly trained,” I expected MAP or VO₂max benchmarks to support that classification. A MAP of 4.35 ± 0.60 W/kg is respectable but not indicative of elite performance. Clarification would help contextualize the generalizability of the findings.
- Lines 120–123: The sequence of exercise tests—with WG1 preceding the GXT—raises concern. It is plausible that fatigue from WG1 affected GXT outcomes, or, conversely, that the 10-minute rest period attenuated the fatigue effects. I would encourage a more explicit rationale for this sequencing.
- Lines 186–189: The statistical interaction in lactate (p = 0.049) is the most prominent group-specific effect. However, the lack of performance effect suggests this difference may not be meaningful. I recommend discussing the possibility of a Type I error or exploring whether this change could have occurred by chance due to multiple comparisons.
- Lines 192: The increase in MAP (3.4%) is reported without significant interaction between groups. I would recommend stating more clearly that this effect occurred independently of the supplement, likely due to continued training.
- Lines 210–217: The CMJ height loss reduction from PRE to POST is modest and not group-specific. I do not believe it supports any supplement-related interpretation and should be discussed as a possible training adaptation.
- Lines 231–244: The opening of the Discussion section rightly emphasizes that this is the first trial of its kind, but I would suggest greater caution in attributing metabolic alterations to the supplement without supporting biomarker data.
- Lines 351–357: The suggestion that recovery between WG1 and GXT “may have allowed reversal” of acetyl accumulation is plausible but speculative. Without measurement of these parameters, such an interpretation should be clearly labeled as hypothetical.
Round 2
Reviewer 1 Report
Comments and Suggestions for Authors
I think all responses to reviewers' comments have been addressed satisfactorily.
I have no comments on the revised manuscript.
Reviewer 2 Report
Comments and Suggestions for Authors
Thank you for incorporating my comments